# Oscillatory Responses to Tactile Stimuli of Different Intensity

**DOI:** 10.3390/s23229286

**Published:** 2023-11-20

**Authors:** Alexander Kuc, Ivan Skorokhodov, Alexey Semirechenko, Guzal Khayrullina, Vladimir Maksimenko, Anton Varlamov, Susanna Gordleeva, Alexander Hramov

**Affiliations:** 1Tactile Communication Research Laboratory, Pushkin State Russian Language Institute, 117485 Moscow, Russia; kuc1995@mail.ru (A.K.); ivskorokhodov@pushkin.institute (I.S.); ansemirechenko@pushkin.institute (A.S.); gkhayrullina@hse.ru (G.K.); maximenkovl@gmail.com (V.M.); gordleeva@neuro.nnov.ru (S.G.); 2Baltic Center for Artificial Intelligence and Neurotechnology, Immanuel Kant Baltic Federal University, 236016 Kaliningrad, Russia; 3Autonomous Non-Profit Organization “Our Sunny World”, 109052 Moscow, Russia; antonvarlamov@gmail.com

**Keywords:** EEG, touch, knismesis, C-tactile afferents, salience

## Abstract

Tactile perception encompasses several submodalities that are realized with distinct sensory subsystems. The processing of those submodalities and their interactions remains understudied. We developed a paradigm consisting of three types of touch tuned in terms of their force and velocity for different submodalities: discriminative touch (haptics), affective touch (C-tactile touch), and knismesis (alerting tickle). Touch was delivered with a high-precision robotic rotary touch stimulation device. A total of 39 healthy individuals participated in the study. EEG cluster analysis revealed a decrease in alpha and beta range (mu-rhythm) as well as theta and delta increase most pronounced to the most salient and fastest type of stimulation. The participants confirmed that slower stimuli targeted to affective touch low-threshold receptors were the most pleasant ones, and less intense stimuli aimed at knismesis were indeed the most ticklish ones, but those sensations did not form an EEG cluster, probably implying their processing involves deeper brain structures that are less accessible with EEG.

## 1. Introduction

Tactile perception has historically been studied as a sensory system that provides information about what comes into contact with the skin, known as mechanoreception. This plays a crucial role in our overall perception of the environment and motor control. This information is processed by aggregating responses from several low-threshold mechanoreceptors (LTMRs) that respond to different types of skin deformation [1]. These receptors are innervated by fast (>15 m/s) A β-type myelinated nerve fibers, allowing for almost immediate central processing.

However, there is another class of touch receptors known as C-LTMRs [2], which are innervated by much slower (>2 m/s) C-type unmyelinated fibers. These C-LTMRs form a distinct tactile perception subsystem, referred to as the C-tactile or CT system. This subsystem selectively responds to slow and gentle touches, often associated with caressing and pleasant sensations. Unlike the discriminative touch system, C-LTMRs project not only to the somatosensory cortex but also to the posterior insular cortex [3], which is involved in affective processing. The CT system is typically seen as a system that rewards prosocial tactile communication in both humans and animals [4]. This is further supported by evidence linking autistic traits to abnormalities in tactile processing [5].

At the same time, the CT system is not the only affective mechanoreception system. Knismesis is a highly alerting and ticklish sensation caused by ultralight movement across the skin, often described as annoying and aversive [6]. It likely evolved as a response to parasitic insects such as mites and mosquitoes [7]. Knismesis is the least studied tactile submodality to date, and its receptors, as well as spinal and cortical processing, remain uncertain. There is a suggestion that it is innervated by fast myelinated fibers, as information about attached parasites must be transmitted rapidly to prompt a reaction. Knismesis may also play a role in touch aversion and defensiveness, both in individuals with autism and those with typical neurodevelopment [8].

Interestingly, two contrasting emotional responses (eagerness to make contact and aversion to contact) can be triggered by quite similar stimuli: slow-moving, light strokes. There may be a mechanism that differentiates these responses, but it is challenging to determine whether it is due to the high specificity of receptors, inhibition at the central nervous system processing level, top-down context-dependent cortical influence, or some other factor. To our knowledge, these two systems have not been previously studied together but rather in comparison to the discriminative touch system. The current research aims to provide a deeper understanding of the interaction between different touch submodalities and the unique characteristics of their central processing.

Therefore, this study hypothesizes that different submodalities (discriminative touch, affective touch, and knismesis) may have distinct neural bases, and investigating brain activity appears to be the optimal approach to unraveling commonalities and differences in their processing.

To test this hypothesis, we compared the neural mechanisms underlying fast, slow, and ultralight touches using Electroencephalography (EEG), a noninvasive method for recording the electric fields produced by neuronal populations in the brain during cognitive processes.

Traditionally, event-related potentials (ERPs) have been the most common way to study perception with EEG [9]. ERPs are waveforms obtained by averaging many stimulus-locked EEG traces. Although tactile ERPs are less common than visual or auditory ones, the core idea remains the same. However, ERPs have two major limitations: first, they usually lack spatial resolution, and second, they provide limited information about EEG features in the frequency domain.

To overcome these limitations, we consider spectral power at different EEG sensors and frequencies as a measure of brain activation. To contrast this multivariate data across conditions, we employ nonparametric statistics based on Monte Carlo randomization. Finally, we associate the differences in EEG power that we uncover with the subjective perception of touches based on tickle and pleasure scores on a visual analog scale (VAS).

## 2. Materials and Methods

### 2.1. Participants

The study included 39 participants (33 females) with normal neurological status. The mean age of the participants was 21.7, SD = 8.52. All participants signed an informed consent form and were instructed that they were free to leave the experiment at any given moment. Most of the sample were participating pro bono out of their own curiosity, and some of the sample received small monetary compensation. Experiments were approved by the Institutional Ethics Committee of the Pushkin State Russian Language Institute (protocol code 17-3-24-118, date of approval 15 July 2022).

### 2.2. Experimental Procedure

The procedure involved the following steps: briefing, recording, answering the questionnaires, and debriefing. During the briefing, a participant was seated in a comfortable office chair, so their left hand was situated on the pillow mount, and their right hand was located on a table surface. A 21” computer display was right in front of them (see Figure 1A). The pillow mount was used to provide comfort for the participants, restrict involuntary arm movements, and compensate for the conical form of the human forearm, ensuring the dorsal forearm lies parallel to the table. A custom-built rotary tactile stimulation (RTS) system (Dancer Design, Ingleton, UK) with two brushes was situated above the pillow mount. A folding screen was used to cover the RTS and their left hand from the participant. After a participant was seated, the lab operators briefed them. During the briefing, a participant learned to use the visual analog scale with a handheld slider. This device was used for subjective assessment of the stimuli in terms of their ticklishness and hedonic traits. Participants received a brief explanation of the experiment procedure and the study goals; all their questions were answered as well. After that, an EEG cap with 32 active AgCl electrodes was applied.

After the EEG cap was ready for recording, an RTS was calibrated. During this stage, it determines the optimal working mode for delivering the desired force. After the calibration, participants were instructed not to move their left hand until the end of the experiment. Then, the participants inserted earplugs to mute the sound of the RTS servomotors. All the further instructions were presented via the display. When the display had no instruction or VAS on screen, a fixation cross was presented. The participants were asked to sit without movement or falling asleep and concentrate on their perception of stroking.

The stimulation included 153 tactile stimuli of three types further referred as fast (brush made of synthetic squirrel fur, force=0.8 N, velocity=30 cm/s), slow (brush made of synthetic squirrel fur, force=0.8 N, velocity=4 cm/s), and ultralight (brush made of synthetic peacock feather, force<0.1 N, velocity=4 cm/s). The stimulation included three blocks for subjective assessment, where after each of the stimuli, the participants were prompted to rate their subjective ticklishness and pleasure with VAS, resulting in 9 assessed stimuli. A total of 144 stimuli were presented without any feedback in a pseudorandom sequence with a random interstimulus interval of 4.5–5.5 s to prevent the time-locking of EEG rhythms. Assessment blocks were in the beginning, in the end, and in the middle of the sequence (See Figure 1B for an experiment timeline).

### 2.3. EEG Acquisition and Preprocessing

We used LiveAmp (Brain Products GmbH) with ActiCap active electrode system. The electrodes were placed according to the international 10–20 system with a modification: electrodes TP9 and TP10, usually placed on mastoids, were placed onto the respective earlobes with a band-aid. The earlobes and forehead were scraped with an alcohol wipe both for disinfection and better conductance. The impedance of all the electrodes was kept at 10 kΩ or below. The EEG was sampled at 500 Hz.

Preprocessing of the recordings was carried out with Brain Vision Analyzer 2 (Brain Products GmbH). The EEG recordings were re-referenced to an average earlobe electrode, filtered in the 0.1–90 Hz range with an additional 49.5–50.5 notch filter (Butterworth filters, 48 dB/octave). Movement artifacts were removed manually. Eye movement, blinking, neck strain, and cardiovascular artifacts were removed with Independent Component Analysis (ICA).

### 2.4. EEG Analysis

EEG recordings were segmented into the epochs time-locked to the onset of the simulation. The lengths of the epoch were 2.5 s for the slow and ultralight stimuli and 0.7 s for the fast stimuli, which is equal to the duration of stimulation. We calculated the wavelet power (WP) [10] for each trial using the FieldTrip toolbox for MATLAB. The Morlet wavelet was used as the basic wavelet function. The frequency range was set from 1 Hz to 40 Hz with a step of 0.25 Hz. The number of cycles denoted as *n*, was dependent on the frequency *f* following the relation n=f.

To reduce variation between participants, we considered normalized wavelet power, NWP, by contrasting post-stimulus WP to the prestimulus WP obtained 2 sec prior to the stimulus onset. The NWP was calculated as follows: $NWP=\left(WP_{\mathrm{post}-\mathrm{stimulus}}-WP_{\mathrm{prestimulus}}\right)/WP_{\mathrm{prestimulus}}$.

### 2.5. Statistical Analysis

For the subjective scores (pleasure and tickle), we assessed the normality of their distributions using the Shapiro–Wilk normality test. Based on the results, we employed the nonparametric Friedman test for non-normally distributed data and repeated measures ANOVA for normally distributed data to examine changes in the scores among fast, slow, and ultralight touches. The statistical analysis was conducted using *JASP*.

To compare NWP across conditions, we employed a statistical F-test in combination with nonparametric cluster-based correction for multiple comparisons and randomization using the Monte Carlo method within the FieldTrip toolbox. The minimum number of neighboring elements required to form a cluster was set to 0, allowing even a single sensor to be considered to be a cluster. The threshold value for the F-statistics was set at 0.0005. We performed 500 permutations for the randomization procedure. The statistical analysis was carried out in MATLAB using the *Fieldtrip* package.

### 2.6. Correlation Analysis

To find an association between the subjective score and NWP, we used repeated measure correlation. Repeated measures correlation (RMC) is a statistical technique used to examine the relationship between two variables when both variables are measured repeatedly on the same individuals. RMC extends the traditional Pearson correlation to account for the within-subject correlation structure [11]. We performed correlation analysis in Python using *pingouin* package.

## 3. Results

Regarding the subjective score of pleasure, we observed that their distributions were normal and showed significant differences across conditions: F(2,70) = 13.748, *p* < 0.001 (ANOVA). Participants assigned the highest score to slow touches (M = 6.88, SD = 1.75), which was significantly higher than the scores for fast touches (M = 5.8, SD = 1.5): t(35) = 4.589, *p* < 0.001 (*t*-test), and ultralight touches (M = 5.4, SD = 2): t(35) = 5.113, *p* < 0.001 (*t*-test). Additionally, there was no significant difference in the pleasure scores between fast and ultralight touches: t(35) = 1.177, *p* = 0.247 (*t*-test) (Figure 2A,B).

The distribution of tickle scores among the respondents was not normal and exhibited significant variation across different types of touch: Chi-Square = 31.393, df = 2, *p* < 0.001 (Friedman test). Participants assigned the highest score to ultralight touches (M = 3.81, SD = 2.23), which was significantly higher than the scores for fast touches (M = 2.08, SD = 2.05): z = 4.242, *p* < 0.001 (Wilcoxon test), and slow touches (M = 2.03, SD = 1.88): z = 4.437, *p* < 0.001 (Wilcoxon test). Furthermore, there was no significant difference in the tickle scores between fast and ultralight touches: z = 0.039, *p* = 0.977 (Wilcoxon test) (Figure 2C,D).

### 3.1. Subjective Attitude

### 3.2. Brain Activity

Comparing the NWP among three conditions (slow, fast, ultralight stimuli), we found four clusters (subsets of EEG channels and frequencies reflecting a significant change in ERSP across conditions). The topograms in Figure 3 display the distribution of F-statistics across EEG sensors at the frequency band of each cluster (indicated below the topogram), while panel E shows NWP spectra (group mean) averaged across EEG sensors of the clusters.

The first cluster with *p* < 0.001 was located in the frequency range 1–5 Hz and included the EEG sensors Fp1, Fp2, F7, F3, Fz, F4, F8, FT9, FC5, FC1, FCz, FC2, FC6, FT10, T7, C3, Cz, C4, T8, CP5, CP1, CP2, CP6, P7, P3, Pz, P4, P8, O1, Oz, O2 (Figure 3A). The NWP in this cluster took the highest and positive value for the fast touches (red line in Figure 3A, right panel) and the lowest value for the slow and ultralight touches (green and purple lines in the Figure 3E).

The second cluster with *p* < 0.001 was located in the frequency range of 8–12.25 Hz and included EEG channels Fp1, Fp2, F7, F3, Fz, F4, F8, FT9, FC5, FC1, FCz, FC2, FC6, FT10, T7, Cz, C4, T8, CP5, CP1, CP2, CP6, P7, P3, Pz, P4, P8, Oz, O2, where the maximal F-value achieved in the right motor area (Figure 3B). The NWP in this cluster took the lowest negative value for the fast touches (red line in Figure 3E). The values for ultralight and slow touches were close to zero (green and purple lines in Figure 3E).

The third cluster with *p* < 0.001 was in the frequency range of 17.25–27.5 Hz and included EEG channels Fp2, F3, Fz, FC1, FCz, FC2, FT10, C3, C4, T8, CP1, CP2, CP6, Pz, P4 (Figure 3C). The NWP in this cluster took the lowest negative value for the fast touches (red line in Figure 3E). NWP for the ultralight and slow touches were around zero (green and purple lines in Figure 3E).

The fourth cluster with *p* = 0.003 is in the frequency range of 28.5–31 Hz and includes EEG sensors channels FC6, C4, T8, and CP6 (Figure 3D). The NWP in this cluster took the lowest negative value for the fast touches (red line in Figure 3E). Again, NWP for the slow and ultralight touches were around zero (green and purple lines in Figure 3E).

### 3.3. Correlation between Subjective Attitude and Brain Activity

Finally, we defined the correlation between the NWP in the revealed clusters and pleasant/tickle scores that participants set to these types of touches. The results are shown in Table 1 for the pleasant score and in Table 2—for the tickle score.

Based on the results, it becomes evident that only the correlation between the tickle score and the NWP in Cluster 3 exhibits a *p*-value below the significance threshold. This particular effect is visualized in Figure 4.

In Figure 4A, the topograms display the distribution of NWP (mean within the group) in the frequency band of 17.25–27.5 Hz, which is associated with the second cluster. Across all types of touches, there is a visible reduction in NWP localized over the motor area, primarily on the right hemisphere. This reduction is most pronounced during fast touches, while in the case of ultralight and slow touches, the decrease is less significant.

Figure 4B, illustrates the correlation between the averaged NWP values and tickle scores. Different colors represent different subjects, and all regression lines exhibit similar slopes across subjects, indicating a consistent linear positive relationship between NWP and tickle scores. It can be observed that higher values of NWP in the right motor area are associated with the sensation of tickling. Conversely, more negative NWP values correspond to lower tickle scores reported by the respondents.

## 4. Discussion

We conducted an experiment in which participants were exposed to three different types of tactile stimulation (fast, slow, and ultralight) applied to their left hand. We assessed their subjective responses to the stimuli in terms of pleasantness and ticklishness scores and recorded neural activation using EEG. The experimental results yielded the following insights: First, participants rated slow touches as the most pleasant, while they gave the highest ticklishness score to the ultralight movements. Second, EEG analysis revealed that fast touches induced the highest positive response in the low-frequency band and the highest negative response across the alpha- and beta-frequency bands in the right motor area. Third, the EEG responses were close to zero for both slow and ultralight touches across all frequency bands. Fourth, we found a low positive correlation between the EEG response in the beta-band across the right motor area and the sensation of ticklishness. Therefore, fast touches, which produced the lowest (negative) response in the beta-band, received the lowest tickle score among respondents.

The subjective assessment of the stimuli was consistent with the initial hypothesis and prior knowledge. The relatively slow and moderately intense touch, similar in those traits to a caressing touch, was expected to be the most pleasant as it was better adjusted to the C-LTMR sensitive range [2]. However, the optimal velocity and force for eliciting knismesis is not yet determined, as most descriptions contain immeasurable descriptions such as “akin to a crawling insect” [6] or “very light movement” [7]. We tried our best to evoke this sensation when designing our ultralight stimuli. The yielded combination of the lowest pleasure score and significantly higher tickle score allows us to say that we were successful in evoking knismesis with a slow-moving artificial peacock feather.

Analysis of wavelet power yielded several clusters of interest that go in line with previous electrophysiological studies of affective touch, albeit scarce. For instance, NWP Cluster 2 demonstrates an alpha suppression over the contralateral motor cortex, a known phenomenon for touch perception [13]. A similar trend was found for the high beta range, resulting in Cluster 3. This pattern of spatial and frequency distribution closely resembles the two-component model of mu-rhythm consisting of 10 Hz and 20 Hz oscillations [14], sometimes called mu alpha and mu beta appropriately [15]. The relationship between the two is debated: whether they are one main beta rhythm with an alpha subharmonic [16], or two distinct rhythms produced by separate generators reflecting different aspects of perception [15,17]. Our findings speak for the latter both in terms of the localization of the clusters and their relation to the perception phenomena. An EEG source reconstruction showed that mu beta is generated at the precentral motor cortex, whereas mu alpha is located at the postcentral somatosensory cortex [14,18]. Although both Cluster 2 and 3 in our study were in similar regions, their topologies slightly differ, with Cluster 2 (8–12.5 Hz) effect epicenter being behind the central line and Cluster 3 (17.25–27.5 Hz) epicenter being more frontal.

The most mu-suppression was found for fast and relatively intense stimuli selected as controls beyond the pressure and velocity range of C-LMRs and yet unknown knismesis mechanoreceptors. Singh et al. showed a larger mu-suppression towards least pleasant stimuli [19]. In this study, the least pleasant texture was a rather harsh, loosely woven textile delivered with a slowly rotating drum. Such a touch may be described as irritating or even borderline painful (unfortunately, Singh et al. did not use the pain scale when gathering feedback) and, therefore, intense. That would explain their results of the most salient stimulus causing the most prominent mu-suppression. Combining their results with ours allows the suggestion that the degree of mu-suppression is primarily defined by the salience of a stimulus. The unpleasantness of a tactile stimulation usually reflects its potential harm to the body or other kinds of immediate danger: excessive force or pressure may lead to a skin injury and other traumas, and the sensation of a moving insect may be a precursor for a bite. For both individual and species survival, potentially dangerous stimuli may have a higher valence. Therefore, we can generalize that in terms of touch perception, unpleasantness and salience would correlate with each other [20] as well as the degree of mu-suppression. Similarly, von Mohr [21] also reported no difference between different types of touch (corresponding to slow and fast stimuli in the current study), with both providing alpha and beta decrease as a response to the physical traits of the stimuli. Our study, however, showed a very smooth response to slow stimulation. This may be explained by the difference in stimulation routine, as our stimuli could not be salient enough to produce such a clustered mu-suppression. However, they were reported as received touch, so they were consciously perceived. That means that while mu-suppression apparently corresponds to the salience of the stimulus, it does not necessarily reflect the act of perception. The correlation of subjective tickle score with mu beta Cluster 3 may further indicate its relation to stimuli salience, as tickle (both laugh-inducing gargalesis and alerting knismesis) is an arousing experience and, therefore, possesses enhanced salience. Cluster 4 (28.5–31 Hz), corresponding to the higher beta range, does not overlap with the conventional mu band range. However, its localization was similar to Clusters 2 and 3, which allows the supposition that several separate, while topographically close oscillators are involved even in a simple act of touch perception.

Interestingly, vicarious touch studies showed results that are similar to ours: visual processing of touch results in mu-suppression as well [22,23] and the effect is more prominent for unpleasant touch perception [24,25]. This could be another piece of evidence for mu-suppression being not just a marker of sensory input but rather a sign of its processing and/or modeling. This can also be viewed through Bayesian brain optics, which considers situation modeling and perception (or model update) as two stages of the same constantly repeated process [26], where stimuli salience matters more than its pure physical traits [27]; however, this lies beyond the scope of the current paper. Theta enhancement (Cluster 1) also goes along with the previous study, which considers theta band to reflect attention processes [28], especially the ones related to sensorimotor integration [29,30,31]. As with the mu clusters, it was the most salient type of stimulation that caused the most prominent theta synchronization. Its spatial traits also resonate well with the previous research, as the most activity was found in the central frontal electrodes. Our Cluster 1 also encompasses delta oscillations, which were hypothesized to be a sign of internal processing [32] and were shown to increase in tasks of active haptic exploration [33]. Thus, it can be viewed as some juxtaposition of those bands, reflecting the general activation of broad perception and attention networks. To ease the understanding of our findings in the current research context, we have prepared a concise comparison (Table 3).

Our findings thus allow us to suggest that cortical response to touch is highly determined by the intensity of the stimulation, with more salient stimulation evoking a plethora of oscillators that integrate this sensation. However, we were unable to find a cluster correlation of affective touch processing, which allows us to suppose that such processes involve deeper brain structures that are harder to localize. This can also be possibly explained by a known inhibitory effect of C-LTMRs afferentation on somatosensory cortex [34]. Those results lay further emphasis on the complexity of the interaction of different tactile subsystems.

Our study has potential limitations. First, impedance values were relatively high. Although we kept them below 10 kΩ, at times, they exceeded 5 kΩ. The latter might reduce the signal-to-noise ratio of EEG signals. To address this limitation, we averaged EEG signals across 50 repetitions for each condition. Second, we utilized surface EEG recordings, which have limited ability to pinpoint the spatial localization of cognitive processes, especially those taking place in deep brain areas. In our study, participants confirmed that slower stimuli targeting affective touch low-threshold receptors were the most pleasant, and less intense stimuli aimed at knismesis were indeed the most ticklish. However, these sensations did not form an EEG cluster, possibly implying that their processing involves deeper brain structures that are less accessible with EEG. Future studies based on source reconstruction of fMRI should confirm this hypothesis.

## 5. Conclusions

A tactile percept encompasses and integrates information from several distinct sensors finetuned to different types of stimulation. Although discriminative tactile perception or haptics received more attention from the researchers, C-afferent system science has left its pioneering phase quite recently, and some other systems, including tickle, are almost neglected. The current study adds to the field of multimodal tactile research in several ways. First, it is one of the first known attempts to quantify the optimal traits for a knismesis eliciting tactile stimuli and deliver it with a precise robotic system. Second, we demonstrated the complexity of networks involved in touch perception that seemingly include several distinct cortical oscillators and some deeper structures that are harder to localize with EEG. Third, we showed a correlation between subjective tickle assessment and wavelet power, which was not studied before, demonstrating that salience is not determined by force and velocity alone. This may add to the development or improvement of a variety of tactile-involved applications, including touch-based interventions and haptic interfaces.

## Figures and Tables

**Figure 1 sensors-23-09286-f001:**
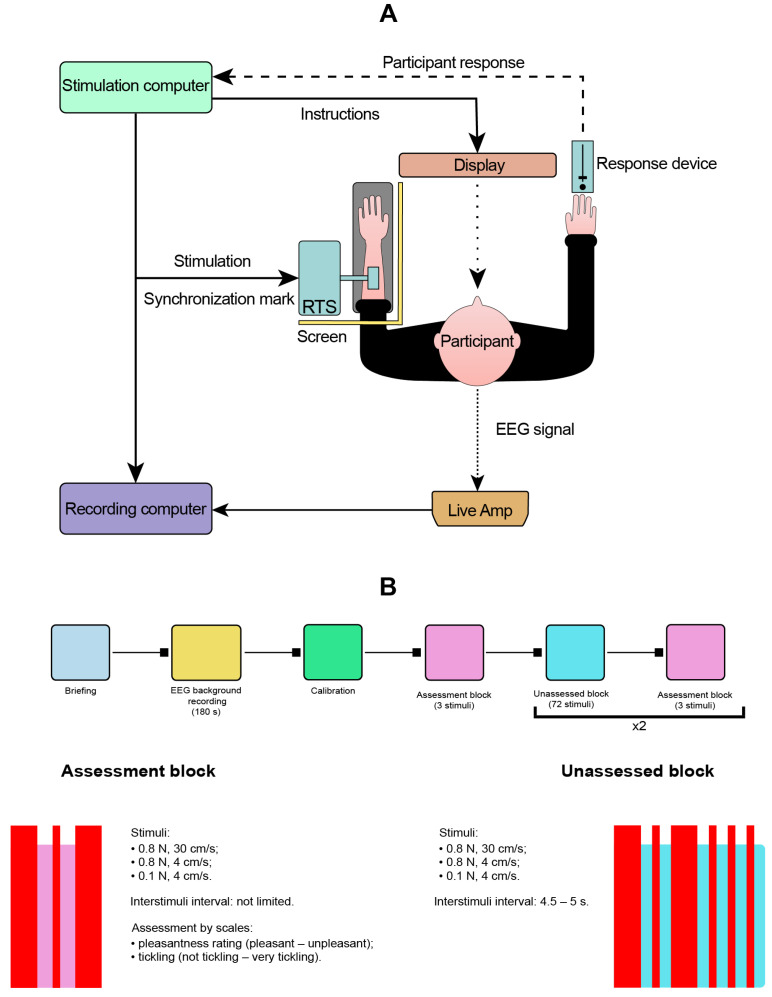
Experimental setup (**A**) and timeline (**B**).

**Figure 2 sensors-23-09286-f002:**
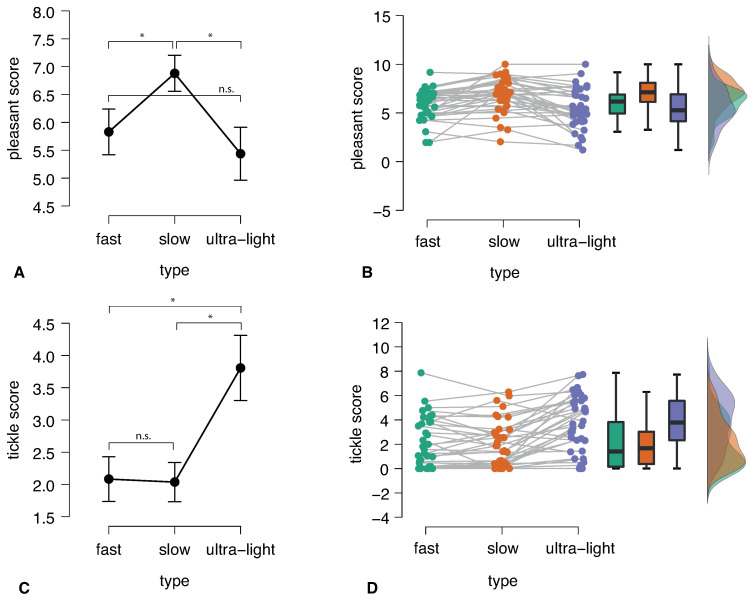
Difference in the pleasant score across conditions: (**A**) group mean and 95%CI, and (**B**) individual trends. Difference in the tickle score across conditions: (**C**) group mean and 95%CI, and (**D**) individual trends. * p<0.05, “n.s.” means not significant. Illustration is generated in JASP (version 0.16.2), an open-source statistics program [12].

**Figure 3 sensors-23-09286-f003:**
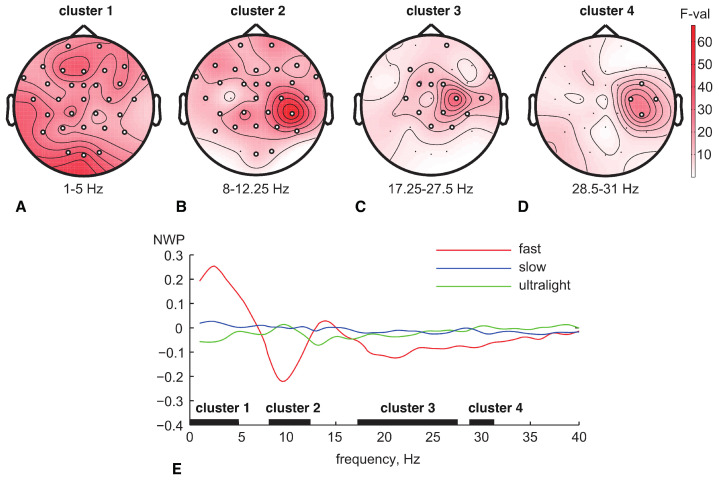
Difference in the NWP across conditions is shown via four clusters (panels (**A**–**D**)). Each panel shows the F-value distribution across sensors and the NWP spectra for three conditions. Panel (**E**) shows the NWP spectra for fast, slow, and ultralight stimuli together with the localization of all clusters.

**Figure 4 sensors-23-09286-f004:**
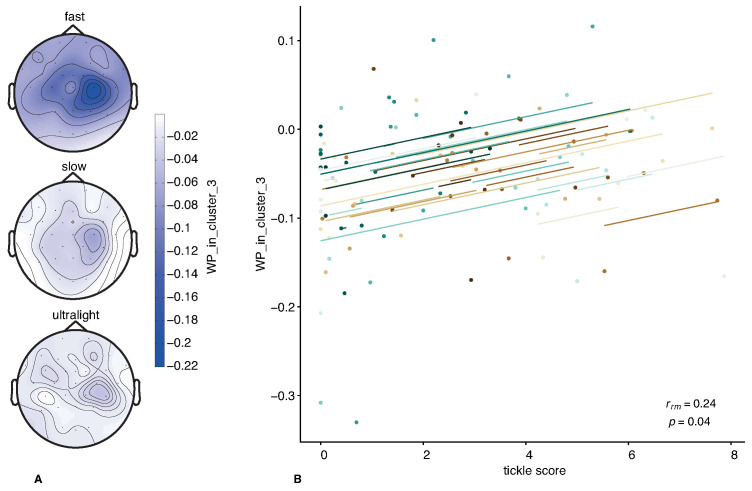
The results of the correlation analysis between the subjective rating (tickle score) and brain activity (NWP in the third cluster). Panel A displays the spatial distribution of NWP across the electrodes in the frequency range corresponding to the second cluster. (data are averaged across all subjects) In Panel B, the colored dots represent data from each participant, while the lines illustrate the slopes estimated for these participants through the repeated measures correlation analysis. Different colors show data of different participants.

**Table 1 sensors-23-09286-t001:** Correlation between the pleasant score and NWP in all clusters.

NWP Cluster	r	Degrees of Freedom	*p*-Value	CI95%
1	−0.19	71	0.101	[−0.406, 0.038]
2	0.14	71	0.238	[−0.093, 0.358]
3	0.05	71	0.686	[−0.184, 0.275]
4	0.08	71	0.493	[−0.151, 0.306]

**Table 2 sensors-23-09286-t002:** Correlation between the tickle score and NWP in all clusters.

NWP Cluster	r	Degrees of Freedom	*p*-Value	CI95%
1	−0.15	71	0.209	[−0.366, 0.084]
2	0.16	71	0.168	[−0.070, 0.379]
3	0.24	71	0.042	[0.009, 0.444]
4	0.12	71	0.320	[−0.115, 0.339]

**Table 3 sensors-23-09286-t003:** Comparison of the brain analysis results with existing literature.

Study	Stimuli	Theta	Alpha	Beta
Current study	Strokes of different forces and velocities delivered with a brush attached to an RTS.	Increase towards the most salient tactile stimuli	Decrease towards the most salient tactile stimuli	Decrease towards the most salient tactile stimuli
Singh et al. [19]	Strokes delivered with a wheel with different fabrics on it.	Not reported	Decrease towards all the stimuli.	Increase towards the most favorable stimuli.
von Mohr et al. [21]	Slow and fast strokes with a watercolor brush delivered by hand.	Decrease towards slow touch, no difference towards fast strokes.	Decrease towards any stimulation, more prominent towards the fast one.	Decrease towards any stimulation, more decrease towards slow strokes in the parietal region.
Schirmer & McGlone [22]	Images of people being touched and matching control images without touching	No difference specific to touch processing.	Decrease towards stimulation, less prominent for touch images.	Decrease towards stimulation, less prominent for touch images.
Peled-Avron et al. [23]	Images of human social touch and control images of inanimate objects as well as human interaction without touching.	Not reported	Decrease towards human touch images as opposed to all the control images.	Not reported
Michail et al. [29]	Painful (laser) and matched non-painful touch (von Frey filaments) stimuli	Increase to both types of stimulation, more prominent towards the painful one.	Decrease to painful stimulation, insignificant increase to touch.	No significant differences.

## Data Availability

Data are available from Ivan Skorokhodov under request (iskor@live.com).

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
