# Peer review of "Oscillatory Responses to Tactile Stimuli of Different Intensity"

_sensors, 2023, doi:10.3390/s23229286_

Round 1
Reviewer 1 Report
Comments and Suggestions for Authors
This is a quite interesting paper about the distinct brain oscillatory responses to tactile stimuli performed by a robotic entity. The paper is well written and quite interesting. The focus, changed from classic ERP to Wavelet power, is an interesting approach, but this and the overall admission of impedance (higher than 5 kilo-ohms) might play against their hypothesis, specially regarding the lack of finding of "a cluster correlate of affective touch processing, which allows us to suppose that such processes involve 304 deeper brain structures that are harder to localize" as they conclude in the last paragraph of the paper and the abstract. I would suggest the authors to comment on this possibility and expand on their explanation of this unability to find such a cluster. I would suggest also to the authors to expand on the description of the cluster related to haptics or discriminative touch. Finally, I have a minor suggestion about figure 2 and is the indication of the statistical significance between the x-axis conditions.
Author Response
We thank the reviewer for reading our manuscript and providing valuable feedback. We are pleased to receive overall positive feedback and have made improvements according to the reviewer’s remarks. We added a paragraph describing limitations caused by the high impedance and surface EEG recordings. Furthermore, we reviewed Figure 2 and provided an indication of statistical significance.
Our study has potential limitations. First, impedance values were relatively high. Although we kept them below 10 k$\Omega$, at times they exceeded 5 k$\Omega$. The latter might reduce the signal-to-noise ratio of EEG signals. To address this limitation, we averaged EEG signals across 50 repetitions for each condition. Second, we utilized surface EEG recordings, which have limited ability to pinpoint the spatial localization of cognitive processes, especially those taking place in deep brain areas. In our study, participants confirmed that slower stimuli targeting affective touch low-threshold receptors were the most pleasant, and less intense stimuli aimed at knismesis were indeed the most ticklish. However, these sensations did not form an EEG cluster, possibly implying that their processing involves deeper brain structures that are less accessible with EEG. Future studies based on source reconstruction of fMRI should confirm this hypothesis.

Reviewer 2 Report
Comments and Suggestions for Authors
1. Some sentences contain grammatical mistakes or are not complete sentences,such as, in line 180, "lowest-" should be "lowest value"; in line 200, "in the Table 2 -" should be "in Table 2".
2. There are some errors in the chart, for example, in Table 1, "0.08," Should be "0.08", the comma should be deleted.
3. The decimal places need to be consistent. Therefore, in Table 2., "0.32" should be "0.320" and "-0.07" should be "-0.070".
4. In line 110 and 116, the manufacturer "(Brain Products GmbH.)" should be written in same format.
5. Too many keywords were listed.
6. Please add a refined Conclusion or take-home message to make the findings and contributions of the paper clearly.
7. What are the advantages of this study compared to other similar experiments? What is the application value of the results?
Comments on the Quality of English Languageno.
Author Response
Q1: Some sentences contain grammatical mistakes or are not complete sentences, such as, in line 180, "lowest-" should be "lowest value"; in line 200, "in the Table 2 -" should be "in Table 2".
Q2: There are some errors in the chart, for example, in Table 1, "0.08," Should be "0.08", the comma should be deleted.
Q3: The decimal places need to be consistent. Therefore, in Table 2., "0.32" should be "0.320" and "-0.07" should be "-0.070".
Q4: In line 110 and 116, the manufacturer "(Brain Products GmbH.)" should be written in same format.
A: We fixed the typos (Q1-Q4) in a revised version of manuscript
Q5: Too many keywords were listed.
A: We reduced the list of keywords
Q6: Please add a refined Conclusion or take-home message to make the findings and contributions of the paper clearly.
Q7: What are the advantages of this study compared to other similar experiments? What is the application value of the results?
A: According to the reviewer’s suggestions (Q6 and Q7), we added Conclusion section containing our thoughts on possible applications of our study as well as a reflection on its novelty.
A tactile percept encompasses and integrate information from several distinct sensors finetuned to different types of stimulation. While discriminative tactile perception or haptics received more attention from the researchers, C-afferent system science has left its pioneering phase quite recently, and some other system, including tickle, are almost neglected. The current study adds to the field of multimodal tactile research in several ways. First, it is one of the first known attempts to quantify the optimal traits for a knismesis eliciting tactile stimuli and deliver it with a precise robotic system. Second, we demonstrated the complexity of networks involved in touch perception that seemingly include several distinct cortical oscillators and some deeper structures that are harder to localize with EEG. Third, we showed a correlation between subjective tickle assessment and wavelet power, which was not studied before, demonstrating that salience is not determined by force and velocity alone. This may add to development or improvement of a variety of tactile-involved applications, including touch-based interventions and haptic interfaces.

Reviewer 3 Report
Comments and Suggestions for Authors
In this study authors investigated sub-modalities of tactile perception using three touch stimuli (fast, slow and ultralight) differing in force and velocity. The study was conducted on healthy subjects who underwent EEG recordings while receiving the tactile stimuli. Participants’ subjective responses to the tactile stimuli in terms of pleasantness and ticklishness scores were assessed.
EEG results revealed a power decrease (suppression) in both the alpha and beta components of the mu rhythm over contralateral sensorimotor areas, and a theta and delta power increase (synchronization) for the fast and intense stimulus, also rated as the most salient by participants.
I think this is a very clear, straightforward and well written manuscript. Results are interesting and well described. Discussion is fluent and pertinent to the main topics of the study. English is fluent and clear. I don’t have any major comments but a couple of very minor suggestions.
1)In the abstract: authors refers to the “the most salient type of stimulation”. I suggest to specify the stimulus they refer to here, to make it more clear for the reader.
2) In the methods section, lines 100-107, authors report that “the stimulation included 153 tactile stimuli of three types…” but then later in the text they report that 144 stimuli were presented without any feedback in a pseudorandom sequence and assessment blocks were taken in the beginning, in the end and in the middle of the sequence. This is a little confusing so I would suggest to better specify the way stimuli were presented and assessment blocks organized in the experimental procedures section.
3) in line 100 “referred as fast slow”: the word slow should be removed. Is this correct?
Author Response
Q1: In the abstract: authors refers to the “the most salient type of stimulation”. I suggest to specify the stimulus they refer to here, to make it more clear for the reader.
A: According to the reviewer’s remark, the type of stimulus mentioned in the abstract was specified.
“EEG cluster analysis revealed a decrease in alpha and beta range (mu rhythm) as well as theta and delta increase most pronounced to the most salient type of stimulation”.
Q2: In the methods section, lines 100-107, authors report that “the stimulation included 153 tactile stimuli of three types…” but then later in the text they report that 144 stimuli were presented without any feedback in a pseudorandom sequence and assessment blocks were taken in the beginning, in the end and in the middle of the sequence. This is a little confusing so I would suggest to better specify the way stimuli were presented and assessment blocks organized in the experimental procedures section.
A: According to the reviewer’s suggestion, we reviewed the corresponding part of the manuscript.
The stimulation included three blocks for subjective assessment, where after each of the stimuli the participants were prompted to rate their subjective ticklishness and pleasure with VAS, resulting in 9 assessed stimuli. 144 stimuli were presented without any feedback in a pseudorandom sequence with a random interstimulus interval of 4.5 – 5.5 seconds to prevent time-locking of EEG rhythms. Assessment blocks were in the beginning, in the end and in the middle of the sequence (See Fig. 1B for an experiment timeline).
We also reviewed Figure 1 and add illustration of the experiment structure.
Q3: in line 100 “referred as fast slow”: the word slow should be removed. Is this correct
A: We have fixed this typo
